# Psychological Correlates of Attitudes toward Pet Relinquishment and of Actual Pet Relinquishment: The Role of Pragmatism and Obligation

**DOI:** 10.3390/ani10010063

**Published:** 2019-12-29

**Authors:** Rita Jacobetty, Diniz Lopes, Jaume Fatjó, Jonathan Bowen, David L. Rodrigues

**Affiliations:** 1Department of Social and Organizational Psychology, Iscte—Instituto Universitário de Lisboa, CIS—Iscte, 1649-026 Lisboa, Portugal; diniz.lopes@iscte-iul.pt (D.L.); dflrs@iscte-iul.pt (D.L.R.); 2Chair Affinity Foundation Animals and Health, Universitat Autònoma de Barcelona, Barcelona Biomedical Research Park, C/ Dr. Aiguader 88, 08003 Barcelona, Spain; jaume.fatjo@uab.cat; 3Queen Mother Hospital for Small Animals, Royal Veterinary College, Hawkshead Lane, North Mymms, Hertfordshire AL9 7TA, UK; jbowen@rvc.ac.uk

**Keywords:** abandonment, attitudes, human-animal relationships, animal welfare, relinquishment, psychometric measures, scale development

## Abstract

**Simple Summary:**

Understanding the psychological correlates of attitudes toward pet relinquishment and actual pet relinquishment is essential to inform interventions, and assess their impact. In this study, we developed new scales to measure attitudes toward pet relinquishment, motives for pet relinquishment, and general trust in pets. With these scales, we showed that attitudes of lack of obligation toward pet relinquishment were more likely in older people, those who perceived their pet as a burden, and those with lower general trust in pets. In addition, we found that attitudes of pragmatism toward pet relinquishment were more likely in men, those who were the main pet caretaker, those who perceived their pet as a burden, those with higher motives for pet relinquishment, and those with lower general trust in pets. Moreover, we found that past pet relinquishment behavior was more likely among people with attitudes of pragmatism toward pet relinquishment. Broadly, these findings advance our knowledge of pet relinquishment, and are likely to inform intervention campaigns to prevent it.

**Abstract:**

Understanding pet relinquishment is essential to inform interventions and assess their impact. In a cross-sectional study, we explored how attitudes of lack of obligation and pragmatism toward pet relinquishment correlated with, and differed according to, sociodemographic characteristics (age, gender, education, political orientation, religion, income, and household), previous animal experience, and owner perceptions of animals (perceiving pet as a burden, motives for pet relinquishment, regret having a pet, and general trust in pets). We adapted and developed three scales to measure attitudes toward pet relinquishment (ATPR), motives for pet relinquishment (MPR), and general trust in pets (GTP), revealing good psychometric qualities. Hierarchical linear regressions showed that attitudes of lack of obligation toward pet relinquishment were stronger in older people, those perceiving their pet as a burden, and those with lower general trust in pets. Attitudes of pragmatism toward pet relinquishment were stronger in men, those who were main pet caretakers, those perceiving their pet as a burden, those with higher motives for pet relinquishment, and those with lower general trust in pets. Furthermore, results showed that past pet relinquishment behavior was predicted by attitudes of pragmatism, but not attitudes of lack of obligation.

## 1. Introduction

Pet relinquishment occurs when caretakers of companion animals voluntarily give up their pets—it is a broad term that includes the behavior of surrendering pets to a third party, abandoning them on their own, or euthanizing them [1]. The relinquishment of owned pets, along with the birth of unwanted litters, fills shelters and municipal pounds with animals and create an overpopulation problem [2,3,4,5]. Interventions to prevent pet relinquishment are few and lack appropriate tools to measure their effectiveness [1,6,7].

In Portugal, data on pet relinquishment is scarce and underreported [8]. The few official reports published by municipal pounds estimate an alarming 22% yearly increase in pet surrenders [9]. Adding to the overpopulation problem, since September 2018 culling became illegal in Portugal, such that municipal pounds are no longer allowed to kill pets as a population control measure [10]. Hence, the population of pets in municipal pounds and animal shelters throughout the country is expected to rise uncontrollably [8]. Efforts to prevent relinquishment in Portugal often include anti-abandonment campaigns, usually rolled out in different media outlets and by different organizations in the beginning of the summer. These campaigns seem to convey a message of responsible ownership and to morally condemn abandonment, possibly aiming to develop stronger attitudes against pet relinquishment as a deterrent of the behavior [11,12,13,14,15,16,17]. Contrary to popular beliefs, however, pet relinquishment is largely stable throughout the year (with the exception of kitten surrenders that are concentrated in the birth season), and holiday-related issues are not frequently reported as reasons for giving up a pet [18] putting into question the timing and angle of the summer campaigns. Furthermore, although studies of efficacy are lacking, these campaigns are likely not reaching their goals of preventing pet abandonment, because the number of surrenders to Portuguese municipal pounds and shelters is still increasing rather than decreasing [19]. Following this, we hypothesize that attitudes might not be changing toward a responsible view on pet ownership, or that these attitudes are not preventing actual pet relinquishment behavior. Hence, we took a step back with the aim of understanding pet relinquishment attitudes to a greater extent, and how these attitudes are correlated with actual relinquishment behavior.

Research on pet relinquishment seems to be gradually moving toward a social and psychological perspective on the issue. This framing is likely a reflection of the greater importance that the human-animal relationship is assuming in our society, and the value of understanding human factors (rather than animal ones) to explain failed human-animal relationships [20,21,22]. Research shows that lacking knowledge about pets—often related with the lack of previous experience in pet caretaking—is among the most reliable predictors of pet relinquishment [2,23,24]. Indeed, people with knowledge deficits about pets tend to develop unrealistic expectations about pet ownership and typical animal behaviors, and tend to be less able to solve problems that arise from their pet’s behavior [20,23,25,26]. Unrealistic expectations can also lead to relinquishment when people expect the relationship with the pet to stabilize after adoption much quicker than it does [27], to expect the amount of work it takes to care for a pet to be less than it really is [28,29], and to expect the pet to play an unreasonable role in the family—e.g., expecting the pet to keep children busy or to teach them sensitivity and nurturance [23]. Changes of lifestyle during pet ownership can also be a risk factor for relinquishment, be it the addition of new members to the household [26], or moving, housing restrictions, and landlord issues (which are reported as one of the top pet relinquishment reasons [3,30,31,32,33,34]). The lack of pet care and investment in the relationship is another variable identified as a predictor of pet relinquishment, as it concerns failing to seek veterinary services, behavior classes, or other forms of professional advice [28,29,32].

Research also shows the importance of contextual and individual differences to understand pet relinquishment. For example, the composition of the household has long been discussed as an important factor, but often with contradictory results. Marinelli et al. [35] found that the level of care for the dog decreases if the household is more than one person, or more than one dog. In contrast, Meyer and Forkman [36] found that the emotional closeness of the owner to the dog was positively associated with owning more than one dog. Having children in the household has been shown to decrease owner’s attachment to their pet [35], and is often a source of incompatibility with the pet’s behavior [27], and correlates with relinquishment behavior [23,32]. Regarding individual variables, research showed that men are more likely to relinquish their pets [2,23], but Salman et al. [32] found that women relinquish cats more often. Also, there is a lower incidence of pet relinquishment among older people [2,23,37], and among those with higher education levels [2]. Income or economic status are not associated with differences in the human-animal relationship [35], but lower income has been associated with greater risk of pet relinquishment [28,29]. Concerning religion, a study in Taiwan found a link between different faiths and relinquishment, with Christians (Western religions) more likely to admit having relinquished a dog, than atheists or followers of traditional Asian religions [38]. Overall, human-animal studies have commented on religion’s conservatism being associated with a more utilitarian treatment of animals, which is rooted in a sense of lesser moral value attributed to non-human animals, mainly in the Judaic/Christian beliefs [39,40]. General conservatism as a political orientation has not been linked to pet relinquishment, but it does impact the moral judgment of animals [41] by considering animals as less moral than humans [42]. Furthermore, animal rights activists do tend to be politically liberal [43].

The level of trust in animals is another variable of interest in human-animal studies. Trust in animals was essential for the domestication of pets [44] and continues to be an important basis of human-animal relationships [45]. Animals are expected to provide companionship and humans to provide care in a cooperative relationship, where mutual trust implies shared risks and rewards, and the break of this trust is the break of the relationship [46]. In human–human intimate relationships, trust is extremely important for the success of the relationship, and is dependent upon previous knowledge of the other and previous expectations of their behavior [47,48,49]. To the extent that successful human-animal relationships are based on trust [50], it is likely that trust in pets is a key variable reliably associated with pet relinquishment.

Despite the growing body of research on pet relinquishment, studies on attitudes toward this behavior are still lacking. Studying attitudes is important to understand pet relinquishment, as the theory of planned behavior [51] reliably shows that attitudes predict behavioral intentions and these, in turn, predict actual behavior in different domains such as condom use [52], smoking [53], and alcohol consumption [54]. In human-animal relationships, several studies have also supported this theory in different attitude–behavior correlations such as responsible pet ownership practices [55,56], dog obesity treatment [57], and sterilization practices [58].

To our knowledge, there are only two studies focusing on attitudes toward pet relinquishment, and both were aimed at developing reliable measures to inform and evaluate the efficacy of pet relinquishment interventions. These studies are particularly relevant because they were conducted in countries with strong ties to Portugal, where people’s perception of pets is likely to be similar. In Brazil, Baquero et al. [59] conducted a population-based questionnaire in an urban area to characterize public opinions and attitudes toward pet relinquishment in hypothetical problem situations. Participants were asked to consider different problematic behaviors (e.g., destroying things, biting, being very disobedient, etc.), select what they believed the owner would do concerning the pet’s fate (e.g., abandon the pet, give the pet to someone else, try to find a solution, or other), and to answer whether they had any reason to abandon their own pet. Results showed that only 9.6% of the participants expressed possible reasons to abandon their own pet, but over-estimated that other people would. Indeed, when asked about what was the fate of other people’s pets, abandonment was the most frequent answer (16% for cats, 17.3% for dogs, and 39.3% for puppies and kittens). The study also found that most people who considered pet abandonment as a possible outcome were not pet owners. According to the authors, this could be explained by pet owners having more positive attitudes toward pets. For example, pet owners should have higher tolerance toward animals and be more likely to avoid relinquishment. One limitation of this study, however, was that participants did not indicate if they had relinquished a pet in the past. Consequently, the authors were unable to examine whether or not attitudes toward pet relinquishment were associated with the likelihood of actual pet relinquishment behavior.

The other study was conducted in Spain by Mazas et al. [60], and aimed at informing educational and training strategies in the development of positive and respectful attitudes toward animal welfare. The authors developed a scale to monitor attitude changes throughout time, or after a specific intervention. The attitudes-toward-animal-welfare (AWA) scale has four components: (1) Animal abuse for pleasure or due to ignorance, assessing attitudes toward certain type of treatment given to animals, e.g., “I sometimes have fun chasing animals.” (2) Leisure with animals, assessing attitudes toward the use of animals in shows, festivals or other recreational activities, e.g., “Bulls are brave animals; their goal is to die in bullrings.” (3) Farm animals, assessing attitudes toward ways of life of animals on farms, during transport and at slaughterhouses, e.g., “Farm animals should be kept in cages so that they can be easily managed.” (4) Animal abandonment, assessing attitudes toward abandoning animals under one’s care, with special reference to pets, e.g., “Abandoned animals feel free.” Results with a sample of 1007 Spanish adolescents and young adults (12–25 years old) showed that women and more educated people had more positive attitudes toward animal welfare (i.e., lower scores on all components of the scale). Moreover, a comparison between the four components showed that participants had the lowest scores on the animal abuse for pleasure and due to ignorance component, and on the animal abandonment component. In other words, participants had more positive attitudes toward preventing animal abuse and animal abandonment.

Studying attitudes may be helpful to explain the complexity of pet relinquishment behavior and to develop new tools to advance this field of research. The aims of this article were threefold. First, as a foundation to our study, we aimed to build and validate a set of scales that reliably assess general trust in pets, motives for relinquishment and owner perceptions of their pet, and attitudes toward pet relinquishment—three important psychological constructs that had been understudied in this field of research. Secondly, we aimed to analyze correlates of attitudes toward pet relinquishment. Specifically, we examined the extent to which psychological variables—such as attitudes toward pet relinquishment, general trust in pets, motives for relinquishment, and owner perceptions of their pet—added explanatory power over the demographic variables—such as gender, education, income, age, political orientation, religion, and household composition. Thirdly, we aimed to show that attitudes are essential to understand actual pet relinquishment behavior by testing the correlation between attitudes and relinquishment behavior.

## 2. Materials and Methods

### 2.1. Procedure and Measures

This study was conducted in agreement with the Ethics Guidelines issued by Iscte—Instituto Universitário de Lisboa. Participants were recruited via public posts on Facebook, through the researchers’ social networks, and via e-mail through different mailing lists (e.g., personal research team members’ mailing lists). They were asked to voluntarily participate in an online questionnaire administered in Portuguese, concerning an extensive study on pets and their owners in Portugal. Clicking on the provided hyperlink, people were redirected to a webpage hosted on Qualtrics, and presented with a detailed description of the study and the respective ethical considerations (i.e., confidentiality and possibility to withdraw without responses being recorded). After agreeing to participate, participants proceeded to the questionnaire. In the first part of the questionnaire, participants provided demographic information (e.g., gender, age). Afterwards, they were presented with the main measures (e.g., experience with animals, attitudes and motives for relinquishment, and owner perceptions of animals). At the end, all participants were debriefed and provided with contact information of the research team. The specific measures used in the analyses presented in this article were as follows.

#### 2.1.1. Demographics

At the onset of the questionnaire, participants were asked demographic information, including age, gender (1 = male, 2 = female), education (1 = basic school, 2 = secondary school, 3 = undergraduate, 4 = MSc/PhD), income (1 = < 580 €, 2 = 581–999 €, 3 = 1000–1999 €, 4 = 2000–4999 €, 5 = > 5000 €), political orientation (1 = with political orientation, 2 = without political orientation), religion (1 = no religion, 2 = religion), living together/alone (1 = living alone, 2 = shared house with family, friends, and partner), and number of children (see Appendix A for a full description of the Portuguese questionnaire and the English meaning and references table for the main items).

#### 2.1.2. Experience with Animals

A second group of questions assessed the respondents’ experience with animals, namely if the participant was the main caretaker of the animal (1 = yes, 2 = no), if they had family or friends with animals (1 = no, 2 = yes), if they worked with animals (1 = yes, 2 = no), who decided to have a pet (1 = themselves, 2 = family), and if the participant did any type of voluntary work with animals (1 = yes, 2 = no) (see Appendix A for a full description of the questionnaire).

#### 2.1.3. Motives for Relinquishment and Owner Perceptions of Animals

A third group of questions concerned the respondents’ motives for relinquishment and owners’ perceptions of animals. Participants were asked if they considered their pet as a burden (1 = never to 7 = always), and if they ever regretted having their pet (1 = yes, 2 = no). Participants were also asked if they trusted pets, by adapting a measure developed by Yamagishi and Yamagishi (1994). This 6-item general-trust-in-pets (GTP) scale assesses trust in pets (e.g., “Generally, pets are honest.” “Generally, pets are trustworthy.”) using 7-point Likert-type scale (from 1 = totally disagree to 7 = totally agree). Moreover, participants were asked to indicate their agreement (from 1 = totally disagree to 7 = totally agree) with a list of motives that would lead them to relinquish their pet. The list of 11 motives—motives-for-pet-relinquishment (MPR) scale—was adapted from the main reasons for relinquishment reported by Salman et al. [32], and included human-related motives (e.g., being allergic to the pet; incompatibility with the pet’s behavior; incompatibility with the pet’s personality) and animal related motives (e.g., expensive pet disease; the pet being old; the pet displaying aggressive behaviors) (see Appendix A for a full description of the questionnaire).

#### 2.1.4. Attitudes toward Pet Relinquishment and Actual Pet Relinquishment Behavior (Criterion Variables)

Regarding our criterion variables, participants were asked to indicate their attitudes toward pet relinquishment (ATPR) and actual pet relinquishment behavior. The 19 items of the ATPR scale included the seven items from the animal abandonment subscale (e.g., abandoning an animal is an irresponsible practice; I would never abandon my pet; animals must be protected by law) of the AWA scale [60], and the remaining 12 items were developed by the authors (e.g., it’s irresponsible to keep a pet that doesn’t adjust to us; there are family circumstances that force the pet’s relinquishment; sometimes there is nothing that can be done to keep the pet). All items were measured with a 7-point Likert-type scale (from 1 = totally disagree to 7 = totally agree). Actual pet relinquishment was measured by asking participants if they had ever relinquished a pet (1 = no, 2 = yes) (see Appendix A for a full description of the questionnaire).

### 2.2. Data Analyses Strategy

We first determined the construct validity and reliability of the ATPR, MPR, and GTP scales. We conducted factorial analyses with principal axis factoring (PAF) and promax rotation [61,62], and determined the internal reliability of the extracted factors using Cronbach’s alpha coefficients. Then, we computed hierarchical regression models to determine the contribution of different groups of variables in the prediction of both attitudes toward pet relinquishment and actual pet relinquishment. These groups of variables were entered sequentially in the models to analyze their specific contribution to the incrementation of the explained variance of the criterion variables. This represented a strong test of their predictive power. Specifically, and regarding attitudes toward pet relinquishment, demographic variables were entered in the first block, followed by a second block of variables relating to experience with animals. In the final block, motives for pet relinquishment were added. Concerning actual pet relinquishment, the significant correlates resulting from the previous regression models were entered in a logistic regression as a first block, followed by a second block of attitudes toward pet relinquishment. In both regression models, and at each step, differences in the total explained variance by successively adding blocks of predictors were analyzed. Also, variance inflation factors (VIF) were calculated for every step of the linear regressions, to account for possible multicollinearity between predictors [63].

## 3. Results

### 3.1. Participants

A total of 700 questionnaire responses were obtained from a convenience sample of Portuguese-speaking participants. After excluding incomplete responses, 504 valid questionnaires were retained for analyses. Most respondents were women (80.2%) and the mean respondent age was 36.30 (*SD* = 11.74; 29 participants did not reveal their age). Regarding education, the majority of the respondents had completed undergraduate studies (50.3%), with 26.7% holding Master or PhD degrees, and 22.2% completed basic or secondary school (39 respondents did not reveal their level of education). In terms of monthly income, 22.7% of participants reported earning less than 580 €, 27.6% between 581 € and 999 €, 38.1% between 1000 € and 1999 €, and 9.3% more than 2000 €. Most participants (83.9%) indicated to live with another person (parents, partner, friends) whereas the remaining lived alone. Of all the participants, 74.4% revealed that they had one child and 25.6% had two or more children. The majority of the participants reported that they did not have any political orientation (50.5%), whereas 29.1% of respondents positioned themselves as left wing, 10.3% positioned themselves as center, 8.4% as right wing, and 1.7% as extreme right. About 57% of participants reported not having any religion, and 35.4% of religious participants defined themselves as Catholic. Almost all participants reported having at least one pet throughout their lifespan (97%) and 85.5% of all participants reported currently owning a pet. Among these latter respondents, 65% reported having dogs, 50% cats, 12.1% small mammals (e.g., hamsters), 14.1% birds, 10.7% fishes, and 9.7% reptiles (some respondents reported owning more than one species). Finally, 85.6% of participants reported no actual pet relinquishment in the past, and 14.4% reported some form of relinquishment (81 participants did not respond to this question). In Table 1, a report is presented of the demographic variables broken-down by actual pet relinquishment in the past.

### 3.2. Construct Validation and Reliability of the ATPR, MPR, and GTP Scales

The final factorial solution of the ATPR scale yielded a two-factor structure retaining seven of the original 19 items (items retained with factor loadings > 0.30; KMO = 0.68) that explains 57.40% of the total variance (factor 1 = 32.84%; factor 2 = 24.56%). These two factors were retained through analyses of scree-plot (factor 1 eigenvalue = 2.30; factor 2 eigenvalue = 1.72). Factorial loadings ranged between 0.73 and 0.44 for factor 1, and between 0.75 and 0.44 for factor 2. Reliability coefficients of both factors yielded moderate results (α_factor 1_ = 0.70; α_factor 2_ = 0.67). Factor 1, designated as pragmatism (the action of relinquishment as a practical consequence [64] to the theoretical problems in the human-animal relationship) toward pet relinquishment, is composed of three items: (i) “It’s irresponsible to keep a pet that doesn’t adjust to us.” (ii) “There are family circumstances that force the pet’s relinquishment.” (iii) “Sometimes there is nothing that can be done to keep the pet.” Factor 2, designated as lack of obligation toward pet relinquishment, is composed by four items. All items are reverse-coded: (i) “I would do anything to avoid relinquishing my animal.” (ii) “Abandoning an animal is an irresponsible practice.” (iii) “I would never abandon my pet.” (iv) “Animals must be protected by law.”

We also checked for differences in the mean scores in each factor of the ATPR scale. An ANOVA revealed that participants globally evaluated each factor differently, *F*(1503) = 897.90; *p* < 0.001, *η_p_*^2^ = 0.64, such that participants agreed more with attitudes of pragmatism toward pet relinquishment factor (*M* = 3.36, *SD* = 1.45) compared to attitudes of lack of obligation toward pet relinquishment factor (*M* = 1.28, *SD* = 0.76). Despite these differences, one-sample *t* tests (*t* test: an indicator of scale sensitivity) against the mid-point of the scale (i.e., value 4) showed that both mean scores were significantly below the mid-point of the scale, both *p* < 0.001.

Regarding the MPR scale, results showed a two-factor structure retaining 10 of the original 11 items (items retained with factor loadings > 0.30; KMO = 0.77) that explains 63.44% of the total variance (factor 1 = 49.47%; factor 2 = 13.97%). These two factors were retained through analyses of scree-plot (factor 1 eigenvalue = 4.95; factor 2 eigenvalue = 1.40). Factorial loadings ranged between 0.93 and 0.50 for factor 1, and between 0.83 and 0.30 for factor 2. Reliability coefficients of both factors yielded moderate to good results (α_factor 1_ = 0.93; α_factor 2_ = 0.69). Factor 1, designated as human-related motives for relinquishment, is composed by five items (“me or family member allergic to the pet”; “pet’s personality incompatible with me”; “pet’s personality incompatible with the family”; “pet’s behavior incompatible with me”; “pet’s behavior incompatible with the family”). Factor 2, designated as animal-related motives for relinquishment, is also composed by five items (“pet’s expensive disease”; “pet’s old age”; “pet’s aggressive behaviors”; “pet’s destructive behaviors”; “urine or feces house soiling”).

As with the previous scale, results from an ANOVA revealed that participants globally evaluated each factor differently, *F*(1422) = 24.91; *p* < 0.001, *η_p_*^2^ = 0.06, such that participants agreed more with human-related motives for relinquishment (*M* = 1.56, *SD* = 1.04), compared to animal-related motives for relinquishment (*M* = 1.35, *SD* = 0.63). One-sample *t* tests against the mid-point of the scale (i.e., value 4) further showed that mean scores in both factors were below the mid-point of the scale, both *p* < 0.001.

Finally, the factor analysis of the GTP scale yielded a one-factor solution retaining all of the original six items (factor loadings ranging between 0.52 and 0.87; KMO = 0.85) that explains 59.9% of the total variance. The internal consistency of the factor was good (α = 0.85). A one-sample *t* test against the mid-point of the scale (i.e., value 4) showed that the mean score in this factor was above the mid-point of the scale, *p* < 0.001 (*M* = 6.24, *SD* = 0.86).

Table 2 presents a summary of the factorial and reliability analyses results for each of these scales.

### 3.3. Correlates of Attitudes Toward Pet Relinquishment

The data analysis strategy previously outlined aimed at investigating the contribution of different groups of variables as correlates of ATPR scores and actual pet relinquishment. More specifically, we examined psychological-level correlates (e.g., motives for pet relinquishment, trust in pets) while controlling for the remaining groups of variables (e.g., experience with animals and sociodemographic characteristics).

Results of the hierarchical linear regression are summarized in Table 3 and Table 4. Regarding the pragmatism toward pet relinquishment attitude, we found that the introduction of the psychological-level predictors—motives toward animal relinquishment, general trust in pets, and perception of their pet as a burden—had the highest increase in the explained variance. As such, the more participants agreed with pragmatism toward pet relinquishment, the more they perceived their pets as a burden; the less they trusted pets in general, the more they agreed with human- and animal-related motives for relinquishment. Also, results showed that this pattern of results was more evident among men and those who were the main caretaker.

Table 4 shows that the introduction of psychological-level predictors contributed to the greatest increase in the explained variance of the lack of obligation toward pet relinquishment attitude. In this sense, participants that perceived their pet as a burden and that distrusted more in pets in general are those who agreed more with a lack of obligation attitude. This pattern of results was also more typical within older participants.

### 3.4. Correlates of Actual Pet Relinquishment Behavior

In order to analyze the predictors of actual pet relinquishment, the significant correlates resulting from the previous regression models were entered in a logistic regression as a first block, followed by a second block of attitudes toward pet relinquishment. This strategy allowed investigating the importance of these attitudes in the prediction of actual pet relinquishment behavior, while controlling for the variables that emerged as significant predictors in the previous analyses (age, gender, being the pet’s main caretaker, viewing the pet as a burden, human-related MPR, animal-related MPR, and general trust in pets). Results of the logistic analysis (see Table 5) showed that the introduction of attitudes toward pet relinquishment in step 2 contributed to an increase in the explained variance of actual pet relinquishment. Specifically, participants with stronger attitudes of pragmatism toward pet relinquishment were more likely to have relinquished their pet in the past. The analysis additionally revealed that men were also more likely to have relinquished their pet.

## 4. Discussion

Our study aimed to understand attitudes toward pet relinquishment and the variables associated with it, and how these attitudes are likely to predict actual pet relinquishment behavior in a Portuguese sample. Results from a cross-sectional study showed that the more people agreed with attitudes of lack of obligation toward pet relinquishment, the more they tended to see their pet as a burden and to generally distrust pets. Arguably, these two perceptions of the pet might hinder the human-animal relationship, are likely associated with a lack of knowledge about animals (i.e., a disability in overcoming the hurdles of pet ownership), and can lead to a divestment of responsibility toward the pet [20,23,25,26]. Results also showed that age was positively associated with lack of obligation attitudes. This goes against the available data from older studies in North America showing that pet relinquishment is less likely for older people [2,23]. Our results may be explained by older generations being less affected by the greater involvement in pet caretaking observed in younger generations, by experiencing lower pet companionship [65] and, the continuity of their parents’ attitudes, an older generation with different cultural norms that did not welcome pets inside their homes [66].

Regarding attitudes of pragmatism toward pet relinquishment, results showed that the more people agree with these attitudes, the more they tend to perceive their pet as a burden, distrust pets and agree with motives for relinquishment (both human and animal related). These three variables point toward the existence of problems in the participant’s relationship with their pets [67], and if a person has experienced difficulties in the human-animal relationship, they are more likely to better understand and sympathize with the circumstances that could lead up to relinquishment, thus viewing the act as justifiable under some circumstances. Being the main caretaker of the pet was also associated with the pragmatism toward pet relinquishment attitude. This is possibly due to owners’ previous knowledge and past experience with pet care that enables them to understand the demands and pitfalls of that responsibility [68] and to recognize that there are circumstances under which people are prone to relinquish their pets. Lastly, our results showed that men have stronger pragmatism attitudes, and this aligns with previous studies that show men to be more likely to relinquish their pets (especially dogs [2,23]).

When we looked at the associations between these two attitudinal components and actual pet relinquishment, we found that agreeing with lack of obligation toward pet relinquishment was not associated with a greater likelihood of actual pet relinquishment behavior in the past. Considering that this factor conveys a sense of responsibility and moral duty toward keeping one’s pet, this finding suggests that appealing to these values might not be an effective strategy to prevent relinquishment. According to Guttman and Ressler [69], appeals to personal responsibility lack efficacy in communication campaigns because instead of empowering individuals by encouraging them toward social transformation, they are an attribution of blame that serves as an obstacle and contributes to stigmatizing those whose behavior the campaigners are trying to impact. Therefore, we argue that a sense of responsibility and moral duty toward keeping one’s pet should not be a central message to convey in anti-abandonment campaigns.

In contrast, agreeing with pragmatism toward pet relinquishment was associated with a greater likelihood of actual pet relinquishment behavior. This finding is possibly explained by the use of justifications to deal with the discomfort of having to relinquish a pet. The effects of the decision to give up one’s pet have long been found to be more complex and legitimate than generally believed, showing that people struggle with making the decision for a long time while tolerating difficult situations and trying to fix them [67]. The cognitive conflict of deciding how to proceed causes them to experience guilt and regret, and once decided, they value the outcome as better than other scenarios and they see it as inevitable and justified [27,67,70]. This finding is particularly relevant because it suggests that actual relinquishment is viewed as a pragmatic and justifiable behavior under some circumstances, as opposed to it just being a consequence of lack of obligation toward the pet.

In addition, in the present article we presented three new scales to measure attitudes toward pet relinquishment (ATPR scale), motives for pet relinquishment (MPR scale), and general trust in pets (GTP scale) with overall good psychometric quality. Our findings showed that the ATPR scale reliably measured attitudes of lack of obligation toward pet relinquishment and attitudes of pragmatism toward pet relinquishment. We also found that the general trust scale used by Yamagishi and Yamagishi (1994) in interpersonal human relationships translated well into human-animal relationships. The GTP scale was a core correlate of attitudes toward pet relinquishment in both the lack of obligation and pragmatism dimensions of the ATPR scale, despite not predicting actual pet relinquishment behavior. The MPR scale was found to be a strong correlate of the pragmatism toward pet relinquishment attitude, particularly in the human-related MPR dimension. Comparing the two dimensions of the scale, the human one explained more variance, better results, and more agreement than the animal-related MPR dimension. Agreeing with motives for pet relinquishment, however, was associated with the pragmatism toward pet relinquishment attitude, but not with the lack of obligation toward pet relinquishment attitude.

Interestingly, results showed that most demographic variables were not associated with attitudes toward pet relinquishment—with the exception of gender and age—, or with actual pet relinquishment in the past—with the exception of gender. Previous experience with animals (or the lack of it), contrary to what was found in older literature [2,23,24], was also weakly or non-significantly associated with attitudes toward pet relinquishment or actual pet relinquishment behavior.

These scales might be further explored to inform the development of a pre-adoption triage tool for shelters and municipal pounds. Assessing the attitudes of adoption candidates toward pet relinquishment (also measuring motives for pet relinquishment and general trust in pets) may be useful in identifying pet relinquishment risk, and therefore reinforcing individual adoption counseling and assistance. These scales can also be useful to pre- and post-test the impact of interventions—such as anti-abandonment campaigns. For these ends, additional validation in representative samples is needed.

### Limitations and Future Research Directions

As a convenience sample of online Portuguese speakers, our findings are not representative of the Portuguese population. Also, due to the recruitment process, our sample derived mostly from the researchers’ extended network that is biased toward people concerned with animal welfare issues, likely providing less participants that have previously relinquished their pet and tending toward a more favorable human-animal relationship. Notwithstanding, the percentage of participants who reported to have relinquished a pet in the past (14.4%) is aligned with previous international literature (14.7% [71], 15.2% [24], and 18.8% [72]). Hence, we argue that our findings are ecologically valid and useful to begin exploring the attitudes of the Portuguese population toward pet relinquishment, and how to help prevent it.

In the different languages of the reviewed literature, the definition of the terms relinquishment and abandonment is not always clear. Although it is agreed by most research that relinquishment is a broader term that encompasses giving up one’s pet through surrender, abandonment, or euthanasia, most Portuguese, Spanish, and Brazilian studies use the term abandonment with this broad meaning. Our approach to this issue in the current study was to adopt the word relinquishment as the broader term, but to also preserve the term abandonment whenever it was the word used in the research materials.

Future research should seek to further explore the ATPR scale, to extend its applicability across different contexts. Samples that include more people who have relinquished a pet in the past should be considered in the recruitment strategy of future studies on pet relinquishment attitudes. Besides previous experience with pets, previous knowledge about pets was identified in the existing literature as a variable of interest in predicting pet relinquishment [2,23,24] and, therefore, developing a measurement to assess it and testing it as a predictor of pet relinquishment attitudes should be explored. Considering that psychological variables were found to be such important correlates of pet relinquishment behavior, further research on this topic should explore other well-established constructs of the social psychology field such as the Investment Model [73], and extending interpersonal relationships processes to the study of human-animal relationships.

## 5. Conclusions

The lack of obligation and pragmatism attitudinal dimensions found in the ATPR scale are two new important psychological constructs to deepen our understanding of pet relinquishment. The ATPR scale, along with the MPR and the GTP scales, has proven to be a valuable measurement in the field of human-animal relationships. Indeed, to our knowledge, the present study is the first to introduce these psychological dimensions to explain pet relinquishment, further advancing scientific knowledge on human-animal relationships and contributing to the void of research regarding pet relinquishment in Portugal.

The goal of this study was to better understand the role of attitudes on pet relinquishment, and our major finding is that pragmatism toward pet relinquishment attitude is correlated with past relinquishment behavior. This attitude represents the agreement with the statements that “it’s irresponsible to keep a pet that doesn’t adjust to us”, that “there are family circumstances that force the pet’s relinquishment”, and that “sometimes there is nothing that can be done to keep the pet”, and these views convey a sense of pragmatism, justification, or even rationalization toward the decision to relinquish, diverging from the notion that people that relinquish are irresponsible, amoral, or maybe rash. On that point, lack of obligation toward pet relinquishment was not associated with a greater likelihood of actual pet relinquishment behavior in the past and, therefore, we should question the use of appeals to responsibility and moral duty to keep one’s pet in anti-abandonment campaigns. Both these findings can be most useful to inform further research, new animal protection policies, and relinquishment prevention interventions.

## Figures and Tables

**Table 1 animals-10-00063-t001:** Actual pet relinquishment in the past by participants’ demographics (in percentages and means).

Sociodemographic Variables	Actual Pet Relinquishment in the Past	
	No	Yes
Gender		
Female	74.3	13.6
Male	11.6	0.5
Education		
Less than secondary school	0.8	0.3
Secondary school	21.9	23.2
Graduation	40.5	6.8
MSc/PhD	22.9	3.8
Income		
≤580 €	20.4	4.2
581–999 €	26.6	3.2
1000–1999 €	32.3	5.7
2000–4999 €	6.2	1.0
≥5000 €	0.5	0
Shared housing/living alone		
Living alone	12.3	2.4
Shared housing	73.3	12.1
Number of children		
0	64.5	10.9
1	12.8	2.1
2	6.6	0.9
3	1.4	0.5
4	0.2	0
Having a pet throughout lifespan		
Yes	85.6	14.4
No	0	0
Political orientation		
Right-wing	7.9	0.5
Centre	8.6	1.7
Left-wing	21.2	5.7
Extreme left	1.5	0.2
No political orientation	86.0	5.9
Religion		
No religion	51.6	10.1
Catholic	34.9	25.5
Age (means)	35.22 (*SD* = 10.83)	34.19 (*SD* = 12.07)

**Table 2 animals-10-00063-t002:** Factorial structures of the attitudes-toward-pet-relinquishment (ATPR), motives-for-pet-relinquishment (MPR), and general-trust-in-pets (GTP) scales.

Items	F1	F2	Corrected Item–Total Correlations
ATPR scale:			
Item 7 (R)	**0.73**	−0.06	0.57
Item 8 (R)	**0.66**	0.03	0.55
Item 6 (R)	**0.66**	−0.09	0.53
Item 14(R)	**0.44**	0.18	0.37
Item 16	0.02	**0.75**	0.54
Item 12	0.01	**0.72**	0.53
Item 9	−0.03	**0.44**	0.37
Eigenvalue	2.3	1.72	–
Explained variance	32.84%	24.56%	–
Cronbach alpha	0.70	0.67	–
MPR scale			
Item 9	**0.92**	−0.02	0.87
Item 8	**0.92**	−0.05	0.84
Item 10	**0.91**	−0.03	0.84
Item 11	**0.90**	0.02	0.87
Item 7	**0.48**	0.24	0.60
Item 5	−0.04	**0.83**	0.66
Item 4	0.09	**0.74**	0.64
Item 6	0.04	**0.55**	0.50
Item 2	0.07	**0.34**	0.35
Item 3	−0.06	**0.30**	0.24
Eigenvalue	4.95	1.4	–
Explained variance	49.47%	13.97%	–
Cronbach alpha	0.93	0.69	–
GTP scale			
Item 2	0.87	–	0.77
Item 3	0.81	–	0.74
Item 1	0.77	–	0.67
Item 6	0.69	–	0.66
Item 5	0.65	–	0.59
Item 4	0.52	–	0.50
Eigenvalue	3.61	–	–
Explained variance	59.90%	–	–
Cronbach alpha	0.85	–	–

(R) = reversed item. Values in bold indicate retained factor loadings.

**Table 3 animals-10-00063-t003:** Correlates of pragmatism toward pet relinquishment attitude (standardized regression coefficients and associated significance).

Outcome	Pragmatism Toward Pet Relinquishment Attitude ^a^		
Correlates	Step 1 ^+^	Step 2 ^++^	Step 3 ^+++^
Sociodemographic			
Age	0.01	0.05	0.03
Gender	−0.18 ***	−0.17 ***	−0.10 *
Education	0.02	0.01	−0.003
Political orientation	−0.03	−0.04	−0.01
Religion	−0.08	−0.09	−0.08
Income	0.04	0.05	0.09
Shared housing/living alone	0.001	0.004	−0.01
Number of children	0.13 **	0.13 *	0.07
Experience with animals			
Voluntary work		0.02	0.002
Work with animals		−0.10 *	−0.09
Family/friends with pets		−0.03	−0.01
Who decided to have animals		0.05	0.05
Pet’s main caretaker		−0.14 **	−0.11 *
Motives for pet relinquishment and owner perceptions of animals			
Pet is a burden			0.11 *
Human-related motives for pet relinquishment			0.24 ***
Animal-related motives for pet relinquishment			0.14 **
Regretted having a pet			0.01
General trust in pets			−0.10 *
Adjusted *R*^2^	0.05	0.07	0.23
Δ*R*^2^	0.06	0.05	0.17
Δ*F*	3.32 ***	2.95 **	16.97 ***

^a^ The higher the score, the higher pet relinquishment is rated as a pragmatic attitude; *** *p* < 0.001, ** *p* < 0.010, * *p* < 0.050; ^+^ Collinearity statistics, as represented by the variance inflation factor (VIF), revealed absence of collinearity between predictors (VIFs ranging from 1.05 to 1.53); ^++^ VIFs ranging from 1.03 to 1.63, revealing absence of collinearity between predictors; ^+++^ VIFs ranging from 1.05 to 1.65, revealing absence of collinearity between predictors.

**Table 4 animals-10-00063-t004:** Correlates of lack of obligation toward pet relinquishment attitude (standardized regression coefficients and associated significance).

Outcome	Lack of Obligation Toward Pet Relinquishment Attitude ^a^		
Predictors	Step 1 ^+^	Step 2 ^++^	Step 3 ^+++^
Sociodemographic			
Age	0.18 **	0.19 **	0.17 **
Gender	−0.09	−0.09	−0.01
Education	0.05	0.05	0.02
Political orientation	−0.07	−0.07	−0.06
Religion	0.01	0.003	0.03
Income	−0.03	−0.02	0.03
Shared housing/living alone	−0.05	−0.05	−0.04
Number of children	0.07	0.07	0.04
Experience with animals			
Voluntary work		−0.02	−0.04
Work with animals		−0.04	−0.01
Family/friends with pets		−0.05	−0.03
Who decided to have animals		0.04	0.02
Pet’s main caretaker		−0.08	−0.05
Motives for pet relinquishment and owner perceptions of animals			
Pet is a burden			0.10 *
Human-related motives for pet relinquishment			0.06
Animal-related motives for pet relinquishment			0.07
Regretted having a pet			0.03
General trust in pets			−0.30 ***
Adjusted *R*^2^	0.05	0.05	0.16
Δ*R*^2^	0.07	0.01	0.12
Δ*F*	3.65 ***	1.004	11.43 ***

^a^ The higher the score, the higher pet relinquishment is rated as a lack of obligation attitude; *** *p* < 0.001, ** *p* < 0.010, * *p* < 0.050; ^+^ Collinearity statistics, as represented by the variance inflation factor (VIF), revealed absence of collinearity between predictors (VIFs ranging from 1.05 to 1.53); ^++^ VIFs ranging from 1.03 to 1.63, revealing absence of collinearity between predictors; ^+++^ VIFs ranging from 1.05 to 1.65, revealing absence of collinearity between predictors.

**Table 5 animals-10-00063-t005:** Correlates of actual pet relinquishment (unstandardized regression coefficients and associated significance).

Outcome	Actual Pet Relinquishment							
	Step 1				Step 2			
Predictors	B	95% CI for Exp B	Wald	Sig.	B	95% CI for Exp B	Wald	Sig.
Empty model	−1.79	–	156.27	<0.001	–	–	–	–
Age	−0.01	[0.97, 1.02]	0.33	0.57	0.00	[0.97, 1.03]	0.02	0.99
Gender	1.60	[1.13, 21.61]	4.48	0.03	1.65	[1.19, 22.80]	4.79	0.03
Pet’s main caretaker	−0.10	[0.39, 2.10]	0.06	0.81	−0.07	[0.40, 2.19]	0.03	0.93
Pet is a burden	−0.05	[0.55, 1.65]	0.04	0.85	−0.09	[0.51, 1.62]	0.10	0.75
Human-related MPR	0.27	[0.94, 1.83]	2.62	0.11	0.08	[0.76, 1.54]	0.17	0.67
Animal-related MPR	−0.16	[0.46, 1.59]	0.26	0.61	−0.28	[0.39, 1.44]	0.74	0.39
General trust in pets	−0.19	[0.58, 1.49]	1.05	0.31	−0.11	[0.61, 1.31]	0.32	0.57
Lack of obligation toward pet abandonment attitude					−0.24	[0.42, 1.47]	0.58	0.45
Pragmatism toward pet abandonment attitude					0.47	[1.27, 2.02]	16.15	<0.001

Nagelkerke R^2^ Step 1 = 0.05; Nagelkerke R^2^ Step 2 = 0.12.

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
