# Peer review of "Psychological Correlates of Attitudes toward Pet Relinquishment and of Actual Pet Relinquishment: The Role of Pragmatism and Obligation"

_animals, 2019, doi:10.3390/ani10010063_

Round 1

Reviewer 1 Report

Dear authors, 

your manuscript provides an interesting and highly relevant contribution to the literature on attitudes on pet abandonment and preventing measures. In general, the manuscript is structured very well and statistical findings are described in a substantive way. Please find some minor comments in teh attached document. 

Reviewer 2 Report

Although the theme is very interesting, the results are not scientifically rigorous to the point of justifying a publication in the journal. In particular, the fact that “attitudes of lack of obligation toward pet abandonment were more likely in people who perceive their pet as a burden, and have lower general trust in pets” is a given and does not add much to my opinion to the important issue of abandonment.

Minor commments:

line 17- 19 (and in several other points) “trust in pets”: this type of terminology should be avoided in a scientific context

Abstract

Line 26 cross-sectional study (N = 504, 80.2% women; Mage = 36.30, SD = 11.74)

These data should be moved to the materials and methods section

Line 81: what relationship? Do he authors refer to the attachment bond?

Line 119: “In Brazil, Baquero, Chiozzotto, Garcia, Amaku an Ferreira”: should be replaced by “Baquero et al.”

Line 135 see above

The text needs to be reviewed by a native English speaker.

Reviewer 3 Report

This manuscript presents some really exciting information about “abandonment” of pets in Portugal.  I have several major concerns which I hope the authors can address.

The first is the terminology of “abandonment”.  I can’t determine what definition of abandonment was used.  In the survey, the authors state that they used relinquishment in one place to indicate a broader term or voluntarily giving up possession of the pet.  In others, they used abandonment.  But in North America, and probably Europe, relinquishment is often used to refer to surrendering a pet to an animal shelter or rescue.  And the references include relinquishment to a shelter, return of an adopted pet to the shelter and abandonment, meaning turning the animal out and not caring for it anymore and seem to all be included—those are very different things.  Please begin by defining what the manuscript is including as abandonment and what term(s) was used in the survey so that it is clear what definition is included in what parts of the manuscript and study.

Second, the survey was administered in Portuguese and parts of it were translated into English from Spanish and part from English into Portuguese.  Translation of surveys into other languages is complex, especially when there may be nuances in the words chosen like abandonment.  Please either add what was done to ensure accurate translation or discuss the limitations and potential impact in the discussion.  In addition, I would really love to see the survey in English for this audience.  That might just be accomplished by making some of the text which lists the questions into tables which clearly state the items…I’ll highlight this below.

This is a convenience sample of online Portuguese speakers.  Please discuss how this might have influenced the results and how generalizable these results are, in the opinion of the authors. This has a huge potential for influencing the interpretation and utility of this study.

Finally, there is some mixing of methods and results sections as well as some missing methods on the statistical analysis component. I’ll point those out as I go along.

Line 44: I think this initial statement is a bit broad and unhelpful.  Are they unwanted or just not yet adopted? Were they strays brought in by someone or surrendered by their owners?  Please revise.

Line 45: here is where some clear definitions of what is and is not covered in the study are needed (the abstracts may also need to be adjusted).  Then the references may need to be checked to ensure they are referring to the type of “abandonment” the study is using.  And surrender to a shelter isn’t the same as returning an adopted pet which isn’t the same as abandoning a pet. 

In the next two paragraphs on page 2, it would be helpful to clarify if these are general statements (if so, reference at least the country) or if the authors are applying them to Portugal.  And please note that some are 15 to 20 years old and may not be accurate even in the populations in which the data were originally collected—there have been a lot of changes in pet keeping in the past 20 years.  I would also like this section to be phrased more as a “what might be happening and why” or “what are the potential concerns” instead of statements of fact.  The references just don’t support these as statements of fact even in the populations studied let alone in other populations or countries or time periods. There are also likely important differences between cats and dogs that would influence this page.  And please edit the reference format errors here and in a few other spots.  

Line 60: this statement needs a reference and a specific geography.

Line 84-91: this is a much better way of explaining what the literature states.

Lin 116: “To our knowledge” should start a new paragraph.  Are these two papers in two other countries likely to be helpful background in Portugal?  Better than work in the US or Canada?  If so, they may be what should be referenced rather than some of the other studies.

Line 153: again, this is a statement and surveys MAY allow us…Or POTENTIALLY explain.  Please edit.

Lines 159-60: in the second aim, I think it would be helpful to briefly add how this will be done to be more in line with the first objective.  Please also note that it is very helpful to keep these 3 objectives in mind throughout the paper, both for the sequence in the methods, results and discussion as well as in headings for different parts of the paper.

Section 2.1 is actual results and should be first section. I would also like to see this summarized in a table, with columns for actually abandoning an animal vs did not and total.  And that 14% of people did this doesn’t seem to be included until the discussion of limitations.

Then Section 2.2 becomes the first methods section.

Page 5 essentially has the survey questions in English with their responses. I would prefer to see this as a table or appendix, depending on the editor and other reviewers. Then it is all easy to read and refer back to.  It would also be helpful to indicate in this survey (and the Portuguese version) which questions were used verbatim from the sources and which were modified or created by the authors.  In the text here there would then be a clearer description including the references of where the questions came from and how they were adapted.  And the footnote would not be needed once the definitions are up front and the survey included in English.

Section 3.1 is methods and should be past tense.   What variables were actually used as the independent variable for the hierarchical models? Please explain here and add to the tables.

Line 267, 271,286, 289, 294, 332: these have reference statistical tests which are not listed in the methods.  Please add there and clearly state what was compared to what and what assumptions for all of the statistical analyses should be checked.  Then in the results, whether or not the assumptions were met can be stated and what was done if the assumptions were not met.

Section 3.2: given the importance of having surveys that actually could be valid and reliable, I might consider making this the first goal in the manuscript and reorganizing accordingly.  And I think that creating some summary tables of the final factors, their loading and other reliability statistics for each scale would be really helpful.  I’m assuming that the one sample t test was an estimate of validity?

Really exciting findings!!

Line 335: be clearer that only variables that were significant at p<0.05 in Step 3 were brought over to this analysis. 

Line 337: I’m not sure this increase is big enough to really highlight.  And there are some concerns about how good R2 are in logistic.  I would also note that the R2 for the linear regression are quite modest and would add a note in the discussion that clearly there are still some other variables to consider when studying this topic.

Table 3: please add or replace the beta with the odds ratios for ease of interpretation.

The discussion will need some revision depending on the definition of abandonment.

Line 353-4: these are older data and a mix of returns and surrender to shelters not true abandonment and in a different country, so I’m not really surprised.

Line 379-81: this is SUCH an important conclusion which could really make some real change.

Line 385-7: this publication is 20 years old, that isn’t recent relative to people’s perceptions of pets, at least in north America.

Paragraphs starting on line 392:  I think that this is an appropriate place to caveat the utility of the surveys and call for some specific additional steps before everyone starts to use them.  Translation, some additional validation etc. would be really good to have!

Line 408-410: again, this seems like an overstatement.  There are only 3 older reference cited and they didn’t really talk about “abandonment” per se. Please rephrase this section.

Conclusion: the importance of this work is the finding about pragmatism and what that means relative to the questions that form the basis for that survey.  Emphasize that and next steps for this type of work.

Line 439-445: I think that 1. There would need to be a lot more work done on the psychometric properties of the surveys and on whether they predict “abandonment” in this context before this could even be suggested. 2. Work in the US has strongly supported the idea that returning an adoption isn’t a bad thing.  And that there are legitimate reasons why people may not be able to keep a pet such as losing housing or jobs or illness which shouldn’t prevent them from getting a pet.  Delete this section.

Round 2

Reviewer 2 Report

Although the authors have responded to the minor comments, my concern about the scientific validity of the results of the work remains such. I am very sorry that I cannot be more positive with the evaluation of this manuscript.